

# The topographic signature of temperature controlled rheological transitions in accretionary prism

Sepideh Pajang[1,2], Laetitia Le Pourhiet[1], and Nadaya Cubas[1]

[1]Institut des Sciences de la Terre Paris, ISTeP UMR 7193, Sorbonne Universite, CNRS-INSU, 75005 Paris, France
[2]Geoscience department, University of Birjand, Birjand, Iran

**Correspondence:** sepideh.pajang@sorbonne-universite.fr

**Abstract.** The local topographic slope of the accretionary prism is often used together with the critical taper theory to determine the effective friction on subduction megathrust. In this context, extremely small topographic slopes associated with extremely low effective basal friction ($\mu \leq 0.05$) can be interpreted either as seismically locked portions of megathrust, which deforms episodically at dynamic slip rates or as a viscously creeping décollement. Existing mechanical models of the long-term evolution of accretionary prism, sandbox models, and numerical simulations alike, generally do not account for heat conservation nor for temperature dependant rheological transitions. Here, we solve for advection-diffusion of heat with imposed constant heat flow at the base of the model domain. This allows the temperature to increase with burial, and therefore to capture how the brittle-ductile transition and dehydration reactions within the décollement affect the dynamic of the accretionary prism and its topography. We investigate the effect of basal heat flow, shear heating, thermal blanketing by sediments, the thickness of the incoming sediments. We find that while reduction of the friction during dewatering reactions result as expected in a flat segment often in the fore-arc, the brittle-ductile transition result unexpectedly in a local increase of topographic slope. We show that this counter-intuitive backproduct of the numerical simulation can be explained and by the onset of internal ductile deformation in between the active thrusts. Our models, therefore, implies significant viscous deformation of sediments above a brittle décollement, at geological rates, and we discuss its consequences in term of interpretation of coupling ratios at subduction megathrust. We also find that, with increasing burial and ductile deformation, the internal brittle deformation tends to be accommodated by backthrusts until the basal temperature becomes sufficient to form a viscous channel, parallel to the décollement, which serves as root to a major splay fault and its back-thrust and delimits a region with small topographic slope. Morphologic resemblances of the brittle-ductile and ductile segments with fore-arc high and fore-arc basins of accretionary active margins respectively allow us to propose an alternative metamorphic origin of the fore-arc crust in this context.

## 1 Introduction

Several studies have suggested a link between the morphology of fore-arc wedges and the seismic behavior of megathrusts, showing a correlation between large subduction earthquakes and fore-arc basins or deep-sea terraces (Wells et al., 2003) or with negative free-air gravity anomalies (Song and Simons, 2003; Wells et al., 2003).





Fore-arc wedges are to the first order well described by the critical taper theory (CTT) (Davis et al., 1983; Dahlen et al.,
1984). This theory assumes that wedges are built by accretion of material equivalent to sand pushed by a moving bulldozer
over a frictional basal décollement. This theory has been very successful in describing the equilibrium morphology of wedges
in response to accretion and as a function of its effective internal and basal frictional strength (Davis et al., 1983; Dahlen et al.,
1984).

The relationship between the fore-arc wedge morphology and the seismogenic behavior has been attributed to spatial vari-
ations of basal shear strength (Song and Simons, 2003) and fore-arc basins are generally associated with an extremely low
effective friction along the seismogenic zone ($\mu \leq 0.05$) (Cubas et al., 2013; Pajang et al., 2021), which is also supported by
heat flow measurements and thermal modeling (Gao and Wang, 2014).

The effective friction $\mu_{eff} = \tau*/(\sigma_n)$ is the ratio of the effective shear stress $\tau*$ to the normal stress $\sigma_n$ acting on a
specific plane. Effective shear stress is the shear stress $\tau$ corrected from isotropic fluid pressure contribution $P_f$ following
$\tau* = \tau - P_f$ (Terzaghi, 1925). Effective friction differs from the internal friction of rocks $\mu = \tau/\sigma_n$, which is constant for
most geological material except mineralogical clays (Byerlee, 1978). Clay minerals are hydrated phyllosilicates, which stability
field is controlled mainly by temperature. Prograde metamorphic reactions that affect clay minerals release water in the system,
which is suspected to raise fluid pressure and diminish effective friction. Clay contents, their nature, and their evolution during
accretion may therefore affect the effective friction of the décollement as a function of temperature history.

Several studies have related the depth-dependence of subduction megathrust seismicity to the diagenetic transformation of
smectite to illite, two clay minerals (Vrolijk, 1990; Hyndman et al., 1995; Oleskevich et al., 1999; Moore and Saffer, 2001).

This transition appears at $\sim 2.5 - 5$ km depth or $100–150^o C$ temperature threshold in clay-rich accretionary complexes (Pytte
and Reynolds, 1988; Hyndman et al., 1995; Oleskevich et al., 1999), and could account for up to 80 percent of the excess in
pore fluid pressure (Bekins et al., 1994; Lanson et al., 2009) necessary to explain the low topographic slope of fore-arcs basins
according to CTT. This transition is found to roughly correlate with the up-dip limit of the seismogenic zone (Oleskevich et al.,
1999). Yet, the relationships between fore-arc basins and seismogenic zones do not work along all subduction zones (Song and
Simons, 2003; Wells et al., 2003).

An alternative explanation for the flat slope of fore-arc basins could be the presence of a weak viscous decollement. This
hypothesis is supported by rock records in exhumed large accretionary complex (Raimbourg et al., 2014; Chen et al., 2018),
which show that the increase in temperature with burial permits to rich the brittle-ductile transition. Yet, this transition from
solid (rate independent or rate weakening) versus fluid (rate hardening) friction is hard to parametrize within the CTT as
it produces rate dependent effective friction and rate dependence is absent of CTT. Long-term models which include the
tectonic and thermal structure inherited from building the fore-arc wedge are thus needed to explore alternative or reconciling
explanations.

The evolution and distribution of long-term internal deformation of fore-arc wedges has been intensively studied by nu-
merical models in two (Burbidge and Braun, 2002; Strayer et al., 2001; Buiter et al., 2016; Stockmal et al., 2007; Miyakawa
et al., 2010; Simpson, 2011; Ruh et al., 2012; Ruh, 2020) and three dimensions (Braun and Yamato, 2010; Ruh et al., 2013;
Ruh, 2016) complementing analogue models (see (Graveleau et al., 2012) for a review). While these models have covered





the influence of many parameters such as the geometry (Dahlen et al., 1984; Davis et al., 1983; Koyi and Vendeville, 2003;
Mandal et al., 1997; Smit et al., 2003; Ruh et al., 2016), basal friction (Colletta et al., 1991; Lallemand et al., 1994; Mulugeta,
1988; Nieuwland et al., 2000; Burbidge and Braun, 2002; Ruh et al., 2012; Cubas et al., 2008), surface processes e.g., (Storti
and McClay, 1995; Mary et al., 2013; Willett, 1999; Leturmy et al., 2000; Konstantinovskaya and Malavieille, 2005; Bonnet
et al., 2007; Fillon et al., 2013; Mary et al., 2013; Stockmal et al., 2007; Hoth et al., 2006; Simpson, 2006), or the presence of
viscous material along the décollement (Gutscher et al., 2001; Costa and Vendeville, 2002; Smit et al., 2003; Couzens-Schultz
et al., 2003; Bonini, 2007; Pichot and Nalpas, 2009; Simpson et al., 2010; Ruh et al., 2012; Yamato et al., 2011; Borderie
et al., 2018). Despite the large amount of published studies, none of them included the dependence of effective basal friction
on temperature due to metamorphic reaction or brittle-ductile transition.

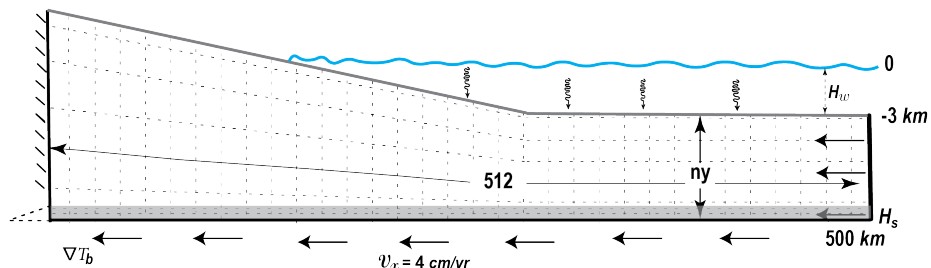

**Figure 1.** Model set-up. Mechanical boundary conditions are a non-deformable backstop on the left side, a constant velocity in the horizontal
direction of $4\,\mathrm{cm.year}^{-1}$, the top boundary behaves as a free surface above sea level while the weight of the water column is prescribed as
stress normal to the deformed boundary below sea level. Temperature is fixed at the surface, the thermal gradient is prescribed across all
other boundaries with the value of zero on vertical ones and value of $\nabla T_b$ at the bottom boundary. The mesh consists of $512 \times$ ny Q2P1
elements which deform to adapt to the deforming top boundary. Parameters are defined in Table 1.

Here, we study how the introduction temperature evolution and its feedback on rock rheology generates deviations from CTT.
For that purpose, we use a typical CTT set up (Figure 1) and we solve for the heat equation on the same domain with a constant
heat flow boundary condition at the base, which corresponds to plate age, to allow the temperature to increase with burial. As
this contribution focuses on the thermal effect, we also include shear heating and thermal blanketing of sediments. We present
different series of 2D thermo-mechanical simulations which assess how the brittle-ductile and the smectite-illite transitions
affect the topographic slope of an accretionary prism and its internal deformation. We briefly discuss internal deformation the
morphology of the wedge and its potential seismic behavior. We, therefore, retrieve the spatial and temporal variation of the
morphologies and deformation patterns and discuss their implications in terms of the fore-arc basin and fore-arc high genesis
and nature.





## 2 Modelling Approach

### 2.1 Method

In order to model the long-term behavior of the accretionary prism, we use pTatin2d (May et al., 2014, 2015), a code based
on finite element method that employs an arbitrary Lagrangian-Eulerian (ALE) discretization together with the material point
method to solve the conservation of momentum (Eq. 1), mass (Eq. 2) and energy (Eq. 3) for an incompressible fluid. It allows
solving thermo-mechanical problems. It has been widely used to model lithospheric scale long-term tectonic problems coupled
to surface processes (Jourdon et al., 2018; Perron et al., 2021) and benchmarked with sandbox experiments (Buiter et al., 2016).

The code solves for velocity $v$ and pressure $P$ assuming conservation of momentum:

$$\nabla \cdot (2\eta\dot{\varepsilon}) - \nabla P = \rho\mathbf{g}, \tag{1}$$

in an incompressible fluid assuming

$$\nabla \cdot \mathbf{v} = 0, \tag{2}$$

of nonlinear effective viscosity $\eta$ and constant density $\rho$. $\dot{\varepsilon}$ is the strain rate tensor and $\mathbf{g}$ the gravity acceleration. Evolution of
temperature $T$ is obtained by solving the time ,$t$, dependent conservation of heat,

$$\frac{\partial T}{\partial t} = \nabla \cdot (\kappa\nabla T) - v\nabla T + \frac{H}{\rho Cp}. \tag{3}$$

The coefficients of eq.3 are $\kappa$ the thermal diffusivity, $Cp$ the heat capacity, and $H$ the heat production. We do not include
radiogenic heat production in our simulation and

$$H = 2\eta\left(\varepsilon_{xx}^2 + 2\varepsilon_{xy}^2 + \varepsilon_{yy}^2\right) \tag{4}$$

corresponds to the sole shear heating.

The Stokes problem eqs. 1 and 2 is solved using high order stable elements ($Q_2$-$P_1$), while the heat equation eq.3 is dis-
cretized on $Q_1$ elements. Physical properties of rocks are computed on Lagrangian markers and projected to gauss points using
constant value per element. Averaging of marker-defined coefficients within element is geometric for viscosity and algebraic
for other properties. At every time step, the surface of the models, $h$, is smoothed according to the Culling diffusive erosion
law,

$$\frac{\partial h}{\partial t} = -\nabla \cdot (k\nabla h) \tag{5}$$

with a diffusion coefficient $k$. Details on implementation of the surface process model in pTatin2d are to be found in (Jourdon
et al., 2018).





## 2.2 Rheological model

We use temperature and pressure-dependent nonlinear rheologies. Effective viscosity is evaluated on material points using first
the Arrhenius flow law for dislocation creep,

$$\eta_{\text{vis}} = A^{-\frac{1}{n}} \left( \dot{\varepsilon}^{\text{II}} \right)^{\frac{1}{n-1}} \exp \left( \frac{Q + PV}{nRT} \right) \tag{6}$$

written in term of the second invariant of the strain rate tensor $\dot{\varepsilon}^{\text{II}}$. The activation volume is set to $V = 8 \times 10^{-6}$ m$^3$.mol$^{-1}$ for
all the lithologies, the other constants A, n, Q are listed for each lithology in Table 1. If the prediction of the second invariant
of stress for a viscous rheology

$$\sigma^{\text{II}} = 2\eta_{\text{vis}}\dot{\varepsilon}^{\text{II}} \tag{7}$$

exceeds the Drucker-Prager frictional plastic yield criterion,

$$\sigma_Y = \sin\phi P + C \cos\phi \tag{8}$$

which depends on $\phi$ the internal friction angle and $C$ the cohesion, the effective viscosity of the marker is corrected in order to
return to the yield envelop with

$$\eta_p = \frac{\sigma_Y}{2\dot{\varepsilon}^{II}}. \tag{9}$$

Finally, the friction angle $\phi$ and cohesion $C$ decrease linearly with accumulation of strain in the plastic regime $\varepsilon_p$ from an
initial friction $\phi_0$ to a final friction $\phi_\infty$ (resp. $C_0$ and $C_\infty$ for cohesion):

$$\phi = \phi_0 - \frac{\varepsilon_p - \varepsilon_{\min}}{\varepsilon_{\max} - \varepsilon_{\min}} (\phi_0 - \phi_\infty), \tag{10}$$

over a range of accumulated plastic strain varying from $\varepsilon_{\min} = 0$ to $\varepsilon_{\max} = 0.5$. This drop of friction and cohesion does not
apply to the décollement. Frictional parameters are listed together with viscous parameters, density, and thermal diffusivity
in Table 1. As all stokes solvers, pTatin2d also applies cut-off values on effective viscosity in order to maintain a reasonably
well-conditioned system of equations. These are set to a minimum value $\eta_{\min} = 10^{16}$ Pa.s and a maximum value $\eta_{max} = 10^{25}$
Pa.s. We made sure that the minimum is never reached to ensure that the frictional properties of the décollement reflect its
extremely low friction.

## 2.3 Initial and Boundary Conditions

The model domain is 500 km long and its initial thickness is either 4 or 7.5 km (Figure 1). It is constituted of 2 layers, a 500 m
thick décollement modeled by shales, while the rest is modeled by sandstones/quartz. The domain is discretized with a mesh of
$512 \times 16$ and $512 \times 24$ Q2 elements, respectively. In the y-direction, two mesh elements are aligned with the initial décollement
layer to better capture its interface and friction at small strain. The décollement material is considered as part of the domain,
as such, we allow shales to be dragged in the rest of the model domain contrarily to frictional boundary conditions adopted





| Parameter | Name | Unit | Sandstone/Quartz[a] | Shale[b] | Sediment/quartz[a] |
|---|---|---|---|---|---|
| A | Pre-exponential factor | $MPa^{-n}.s^{-1}$ | $6.8 \times 10^{-6}$ | $1.3 \times 10^{-67}$ | $6.8 \times 10^{-6}$ |
| n | Exponential stress | - | 3 | 31 | 3 |
| Q | Activation energy | kJ | 156 | 98 | 156 |
| $C_0$ | Initial cohesion | MPa | 2 | 0.1 | 2 |
| $C_\infty$ | Final Cohesion | MPa | 1 | 0.1 | 1 |
| $\phi_0$ | Initial friction | $^o$ | 25 | 5 | 25 |
| $\phi_\infty$ | Final friction | $^o$ | 10 | $5^*$ | 10 |
| $\kappa$ | Heat diffusivity | $m^2.s^{-1}$ | $10^{-6}$ | $10^{-6}$ | $10^{-7}$ |
| $\rho$ | Density | $kg.m^{-3}$ | 2400 | 2400 | 2000 |

**Table 1.** Variable rheological parameters and coefficients for the different lithologies; creep parameters [a] from (Ranalli and Murphy, 1987) and [b] from (Shea and Kronenberg, 1992). * In models **M13** to **M15** the friction in the Shale is temperature dependent.

to benchmark the code with sandbox experiments (Buiter et al., 2016). The shortening of the model is driven by a constant horizontal velocity $v_x = 4$cm/yr applied both at the right and bottom boundaries. Above the décollement level, the left side of the domain is rigid. Within the 2 mesh elements of the left boundary which belongs to the décollement, a vertical velocity gradient is applied to ensure the continuity with the bottom boundary. The surface of the domain is modeled with a free surface

above sea level (located 3 km above the top of the mechanical model, Figure 1), below sea level additional normal stress:

$$\sigma_n = H_w \rho_w g \tag{11}$$

is applied on the deformed surface to mimic the weight of water, yet shear stress is zero like above sea level.

The thermal boundary conditions assign the temperature $T_0 = 0^o C$ at the surface, a constant thermal gradient $\nabla T_b = \left.\frac{\partial T}{\partial y}\right|_{y=y_b}$ at the base and no horizontal gradient/(insulating boundary) on the vertical walls of the domain. As we assume

no radiogenic heat production, the initial temperature in the domain is fixed to:

$$T = T_0 - \nabla T_b y \tag{12}$$

consistently with the boundary conditions.

### 2.4 Post-processing

Models are named by a number within a bullet on the left of the panels. This number refers to Table 2 which contains all the

specific parameters used for this realization of the model. For each simulation, we show the finite strain and the current state and strain rate.

The finite strain figure displays the lithologies of rocks and the accumulated plastic strain on markers with a linear colormap. Markers with values of plastic strain $\varepsilon_p$ larger than 2 or 3 can be interpreted as being part of a fault. In order to better show the deformation, sandstone sequences have been colored with thin layers that have no physical existence. Sediment markers

deposited by surface processes are colored by the time of deposition from brown to yellow colors.





The current state figure displays whether the material is yielding plastically (blue) or deforms viscously (red). It is overlaid by green shades of the second invariant of strain rate to outline structures that are currently active by comparison to finite strain. A cut-off range from $5 \times 10^{-16} s^{-1}$ to $2 \times 10^{-14} s^{-1}$ is used for the post-processing. The actual values span a larger range. We also represent three isotherms (180, 300, and $450^{o}C$) which have been chosen to correspond to the onset of viscous

deformation in low strain islands, brittle-ductile transition for quartz at average strain-rate and completely ductile behavior. In the simulation with dehydration reactions, the $120^{o}C$ isotherm is added to locate the onset of the dehydration reaction. Finally, we represent the local slope with a color code at the top of the slice.

## 2.5 Experimental plan

| Model | $H_s$ (km) | ny | $\nabla T_b$ ($^{o}C/km$) | $k$ | SH | DR |
|---|---|---|---|---|---|---|
| **M0** | 4 | 16 | - | $10^{-6}$ | off | off |
| **M1** | 4 | 16 | 15 | $10^{-6}$ | on | off |
| **M2** | 4 | 16 | 15 | $10^{-6}$ | off | off |
| **M3** | 4 | 16 | 15 | $10^{-5}$ | on | off |
| **M4** | 4 | 16 | 15 | $10^{-5}$ | off | off |
| **M5** | 7.5 | 24 | 15 | $10^{-6}$ | off | off |
| **M6** | 7.5 | 24 | 15 | $10^{-5}$ | off | off |
| **M7** | 7.5 | 24 | 15 | $10^{-6}$ | on | off |
| **M8** | 7.5 | 24 | 15 | $10^{-5}$ | on | off |
| **M9** | 4 | 16 | 25 | $10^{-6}$ | on | off |
| **M10** | 4 | 16 | 25 | $10^{-5}$ | on | off |
| **M11** | 7.5 | 24 | 25 | $10^{-6}$ | on | off |
| **M12** | 7.5 | 24 | 25 | $10^{-5}$ | on | off |
| **M13** | 4 | 16 | 15 | $10^{-6}$ | off | on 5-0.1-10 |
| **M14** | 4 | 16 | 15 | $10^{-6}$ | on | on 5-0.1-10 |
| **M15** | 4 | 16 | 15 | $10^{-6}$ | on | on 5-0.1-5 |

**Table 2.** Variable parameters. Hs= Domain thickness, ny = Number of vertical element, $\nabla T_b$ = basal thermal gradient, $k$ = coefficient of diffusion for surface process model, SH = Shear heating, DR=Dehydration reaction followed by values for temperature dependant internal friction angle in $^{o}$.

Our aim is to provide the community with a first assessment of the topographic expression of the change in thermo-
rheological regime with burial in an accretionary prism. We, therefore, have chosen a simple reference model which produces well defined thrust and back thrust, with a moderate amount of sedimentation. The effects of softening, basal friction, and basal slope on accretion have already been thoroughly studied e.g., (Graveleau et al., 2012; Buiter et al., 2016; Ruh et al., 2012, 2014).



Hence, we here concentrate on parameters that are known to affect the geotherm: basal heat flow (represented by basal
thermal gradient $\nabla T_b$), initial burial ($H_s$), coefficient of diffusion of the topography (erosion and sedimentation) (k), and shear
heating (SH) following the plan listed in Table 2. In all the experiments, thermal blanketing (Jeffreys, 1931; Wangen, 1994) is
very roughly simulated by using a lower thermal diffusivity for sediments produced by the surface process model (see Table
1). As we do not simulate the compaction of sediments with burial, thermal insulation is probably over-estimated but it allows
testing potential effects of sedimentation on the thermal state of the accretionary wedges.

## 3 Main Results

### 3.1 Effect of brittle-ductile transition at one glance

In Figure 2, we first compare the results of our reference model (**M1**), with a 4 km thick pile of sediments affected by moderate
erosion sedimentation, shear heating, and a thermal gradient of $15^o$/km which corresponds to a plate age of 65Ma, with the
same model run without thermal coupling (**M0**).

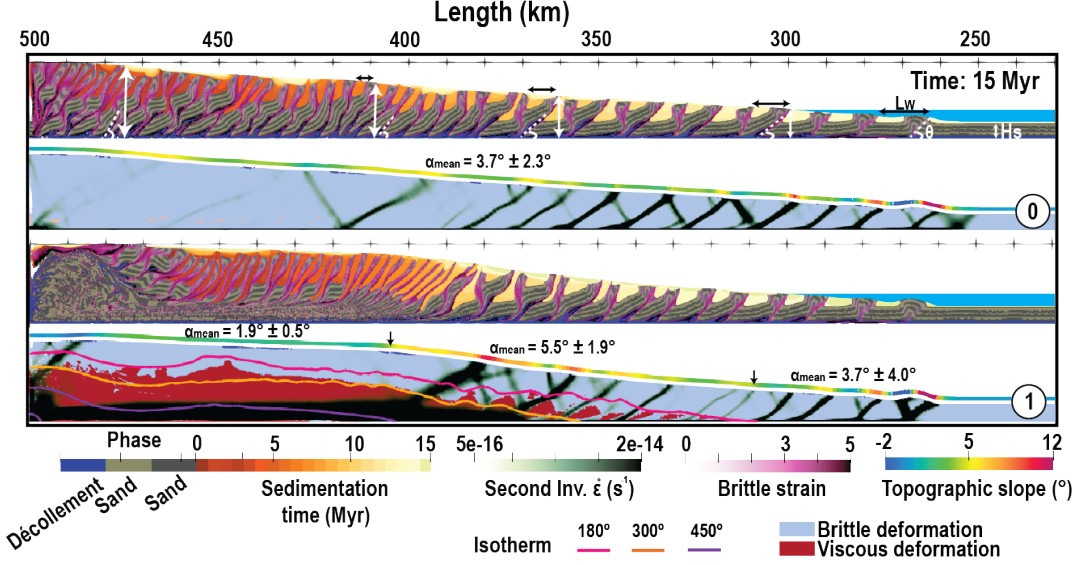

**Figure 2. Model M0**: without temperature compared to **Model M1**: the same experiment with temperature dependant rheologies, both after
15 Myr evolution. The finite strain and the current state and strain rate are shown for basal frictional angle ($\phi_b$=5$^o$) and internal friction angle
dropping from 25$^o$ to 10$^o$ with strain. The yellowish-brown color illustrates syn-tectonic sedimentation in the form of piggyback basins which
represent a given age that corresponds to the time of deposition. Where the material is yielding plastically is in blue or deforms viscously is
in red with the onset of 180, 300, and 450 $^o$C isotherms. Black arrows display decreasing thrust space towards the backstop and white arrows
show increasing sequence thickness and inclination angle $\theta$. The local topographic slope with a color code shown at the top of the slice is
affected by temperature and frictional properties.





While the frontal parts are not significantly different, with similar tapers and ramp spacing, as soon as the thickness of the wedge doubles (close to sea level at $x$ c.a. $300\,\mathrm{km}$), the back-thrusts are more active in the purely brittle wedge than in the thermally controlled wedge for these brittle parameters. This results in the deactivation of one ramp over two, which leads to the formation of the distinct twinned-slice patterns observed between $x = 430$km and $x = 300$km in **M0**. In **M1**, the twinning of slice by back thrust occurs only once the temperature at the base of the model reaches $300^{o}C$.

Looking at the internal part of the two accretionary prisms, the differences become of course more striking. In the case without thermal coupling (**M0**), deformation continues by pairing together more and more slices within sequences of thrust and back-thrust which root deeper and deeper as the accretionary prism thickens. The prolonged activity of these out-of-sequence thrusts and back trust is best measured by the small out-of-sequence basins that form at their top, discordant on the older sediments. Very little exhumation occurs close to the back-stop. In the case of thermal coupling, as soon as the $450^{o}C$

isotherm is reached, deformation becomes highly partitioned vertically. A thick layer at the base accommodates the simple shear and branches on main frontal thrusts which root at the brittle-ductile transition. The ductile material is exhumed along a normal fault that roots on the backstop. In between, the deformation in the brittle part is either very distributed or almost nonexistent as the strain rate remains below our visualization threshold.

     In the end, zooming out of these details and looking at the topographic slope, we can see that while the brittle accretionary

wedge displays a rather constant $4\ ^{o}$ slope, as predicted by the CTT taking into account the softening parameter (Ruh et al., 2014), the mature brittle-ductile wedge forms three distinct segments with a rather low but non zero topographic slope close to the backstop, a CTT predicted slope close to the toe of the wedge and in between a zone with a distinctively larger topographic slope which corresponds to the brittle-ductile transition.

### 3.2    From the emergence of the transition slope to steady state wedge

Figure 3 shows the structural evolution of the reference simulation **M1** through time. The wedge grows horizontally by in-sequence thrusting and vertically by reactivation of thrusts within the wedge. Most of the horizontal shortening is accommodated by the active frontal thrust. As the wedge evolves, the surface slope also changes and we detail here its evolution in time.

     After 1 Myr, accumulated plastic strain shows that the deformation is strongly localized along the frontal part and on the

décollement. Shear bands initially occurred in conjugate sets; with ongoing shortening, landward dipping shear zones are preferred and back-thrusts are almost abandoned. The wedge is trying to reach a critical state by creating a topographic slope. Due to internal frictional softening, the brittle wedge present a slope $\bar{\alpha}$, which corresponds to the slope predicted by CTT for basal friction $\phi_b$ and internal friction of $17^o$, i.e. the average of $\phi_\infty = 10^o$ and $\phi_0 = 25^o$.

     At 5 Myr, some patches of viscous deformation appear in between the main faults at $x = 413km$ once the base of the model

reaches $180^oC$. From that point, landward plastic strain and the second invariant of the strain rate indicate the activity of out-of-sequence thrusts and steep back-thrusts which dissect the former slices and tend to flatten horizontally the sequences as the former ramps become more and more vertical. This phenomena induces vertical thickening that correlates with the occurrence of internal viscous deformation at depth and with an increase in local slope $\bar{\alpha}$ independently of basal friction which remains



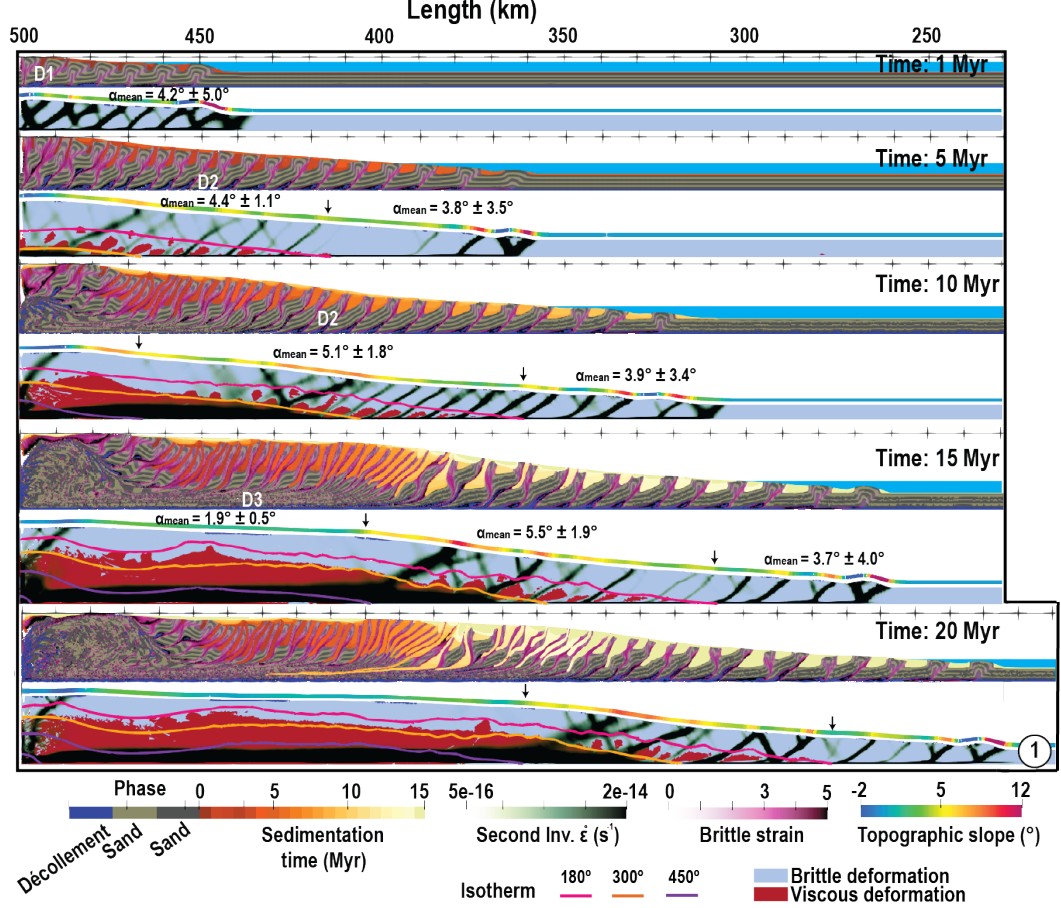

**Figure 3.** Temporal evolution of the reference model (**M1**) from beginning to 20 Myr of shortening.

constant. This increase in slope causes an increase in sedimentation rate at the front where piggyback basins tend to be more

starved than in the initial phase favoring the activity of thrusts, and a lengthening of the slices as predicted by other studies like
Storti and McClay (1995) using sandbox experiments or Simpson et al. (2010) using numerical simulations.

As the shortening goes on, the temperature continues increasing at the back of the wedge due to burial. Between 5 and 10
Myr, the strain rate shows that the thickness of the décollement increases to reach 2 km. A normal fault forms near the backstop
and starts accommodating the exhumation of high metamorphic grade rocks, which deforms in a ductile manner. At the front

of this wide ductile horizontal shear zone, which terminates right at the $300^{o}$C isotherm, roots a system of low angle thrusts
and high angle backthrusts which separates the warm distributed part of the wedge from the thrust dominated section of the
wedge.

From 10 to 15 Myr, the warm part of the wedge grows in length and its topography flattens. The vertical partitioning
between simple shear at the base and pure shear at the top becomes evident. At the front of this ductile wedge, the transition





zone, where thrusts and their back-thrust roots directly in the ductile décollement near the 300$^o$C isotherm, develops giving rise to a distinctively larger slope than predicted by CTT. At the toe of the wedge, the slope remains constant and CTT compatible.

At 20 Myr, the overall architecture of the accretionary prism has not changed, the whole wedge is just shifted towards the right as the ductile part of the wedge has grown wider. One could state that the brittle-ductile wedge has reached some sort of steady state between 10 and 15 Myr. Rocks that are incorporated in the wedge start by rotating along with brittle thrust and

maybe deforming in a pure brittle manner within the slice. In a second semi-brittle phase, internal viscous-ductile deformation affects the whole tectonic slices separated by brittle thrusts. This phase corresponds to crossing the zone with a higher than normal topographic slope. Finally, depending on whether they were incorporated in the ramp or not, they go through passive rotation by distributed pure shear thickening associated with low temperature or intense ductile simple shear before being exhumed for large temperature.

In conclusion, the large slope segment which corresponds to the brittle-ductile transition is acquired as soon as some viscous internal deformation occurs in between faults. The brittle-ductile wedge reaches a steady state when ductile deformation becomes predominant on the décollement that is for temperature greater than 450$^o$C.

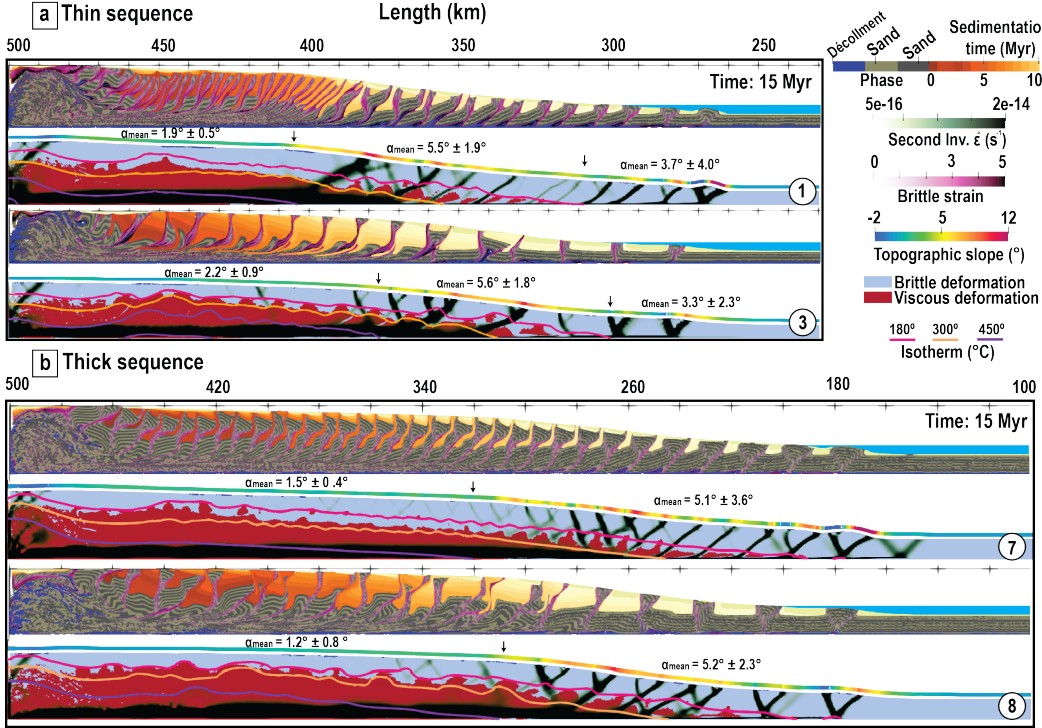

**Figure 4.** Models with shear heating on after 15 Myr of shortening. **a.** Thin and **b.** Thick sequence pile models. **M1**, **M3** and **M7**, **M8** are the same experiment with normal and high sedimentation rate respectively. Experiment parameters are given in Tables 1 and 2 and color code for markers is given in Figure 2.





### 3.3 Sensitivity Analysis

Depending on the chosen thermal parameters, the three segments described above are more or less developed in our simula-
tions. A major player in the development of the ductile flat part of the wedge associated with high grade metamorphic rock
exhumation and formation of a forearc basin is the occurrence of shear heating.

Shear heating might be largely reduced by several factors and noteworthy enough the presence of water which reduces the
ductile strength of the material or by thermal pressurization during earthquakes (Sibson, 1973; Lachenbruch and Sass, 1980;
Mase and Smith, 1984, 1987; Segall and Rice, 2006). Gao and Wang (2014) actually showed that megathrusts that produce
great earthquakes tend to dissipate less heat than megathrusts that slip mainly by creep. Hence, our shear heating models give
a maximum bound for the heat that could be produced in the system. In order to study the other bound, we ran some model
with shear heating off which would correspond to a system where most slip is accommodated by earthquakes.

Results are clear, all models with shear heating (Figure 4) develop a large flat area because the shear at depth helps the
temperature to rise above the $450^oC$ isotherm, which correlates with the formation of the topographic plateau where high
grade metamorphic rocks are exhumed. Models with no shear heating hardly develop a plateau and a normal fault to exhume
high grade material at the back (Figure 5).

Actually, only models with large erosion coefficients, i.e. **M4** and **M6**, do produce exhumation and a small plateau when
shear heating is deactivated. The peak metamorphic temperature of rocks exhumed at the back-stop in presence of shear heating
or in models with large erosion rate is compatible with thermochronometry studies in stationary accretionary prism like Taiwan
(Suppe et al., 1981; Willett and Brandon, 2002) which indicate the samples exhumed to the surface by rock uplift to compensate
for the mass lost by erosion (Fuller et al., 2006) have experienced temperatures in excess of 300–365$^oC$ but below $440^oC$ e.g.,
(Lo and Onstott, 1995; Fuller et al., 2006).

Models with larger sedimentation rate (models **M3**, **M4**, **M6**, **M8**, **M10**, **M12**) produce, as expected from former studies
(Storti and McClay, 1995; Simpson et al., 2010), a smaller number of thrusts and a larger spacing between them. Erosion and
sedimentation also participate at reducing the slope at every step, thus favoring out-of-sequence activity (Figure 5 **M2** and
**M4**). In our models, sediments also affect the thermal regime because they are attributed a lower thermal diffusivity; thick
sediment sequences act therefore as a blanket isolating the heat flux coming from below. As a result, temperature raises faster
at depth when large sedimentary basin forms. This explains why, in absence of shear heating (Figure 5), only models with large
sedimentation rate develop a ductile flat at the back (Figure 5 **M4** and **M6**).

Thermal blanketing also affects the geotherm at a smaller scale as shown by the distinctive wiggles in the isotherms. For
small basins, these wiggles are limited to the $180^oC$ isotherm (Figure 5 **M2** and **M5**), but for large basins the $300^oC$ isotherm
is affected at larger wavelength and potentially feeds back on the rheology (Figure 5 **M4** and **M6**).

In absence of heat production and large vertical advective terms, the temperature is more or less proportional to depth and
thermal gradient in the models, experiments with thick sequences (**M5** and **M6** in Figure 5 and **M7** and **M8** in Figure 4) or
larger imposed basal gradient (**M9, 10, 11, 12** in Figure 6) reach the onset of brittle-ductile transition earlier. As a result, the
completely brittle part of the accretionary prism, located at the toe of the wedge, is less developed in models with a larger pile


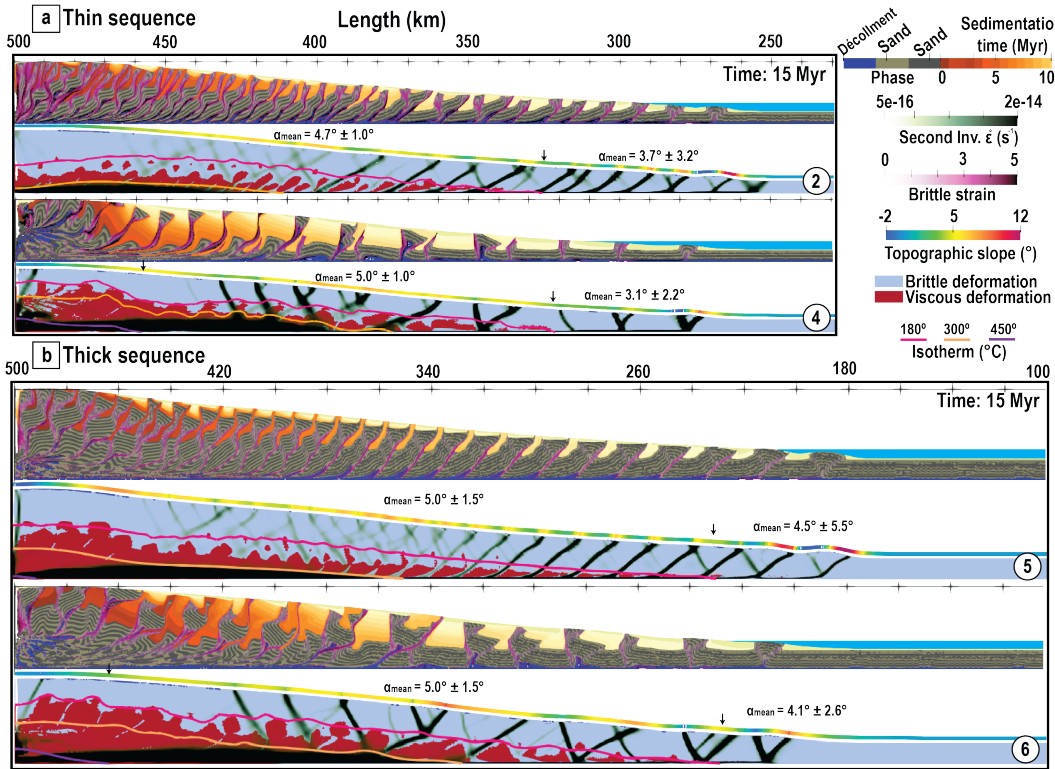

**Figure 5.** Models with no Shear heating after 15 Myr of shortening for **a.** Thin and **b.** Thick sequence pile models. Normal sedimentation rate use in **M2** and **M5**, and high erosion rate in **M4** and **M6**. Refer to Figure 2 for color codes and Tables 1 and 2 for experiment parameters.

of incoming sediments and in models with a larger thermal gradient. The local slope in these models is always larger than the CTT predicted one and back-thrust appear earlier in the history of deformation.

### 3.4 Effect of Dehydration reactions

We now report some models which intend to tackle the effect of fluid over pressure due to dehydration of shale materials which potentially corresponds to smectite-illite transition. In clay-rich accretionary complexes, this transition appears at $\sim 2.5 - 5$ km depth corresponding approximately to 100–150$^o C$ (Pytte and Reynolds, 1988; Hyndman et al., 1995; Oleskevich et al., 1999). Smectite and illite clays are frictionally weak but illite is slightly stronger (Morrow et al., 1982; Saffer and Marone, 2003). We, therefore, consider that our clay rich décollement is initially smectite rich with an internal friction angle of 5$^o$. Once

dehydration reaction is terminated the same décollement is considered illite rich and affected a friction angle of 10$^o$. We do not have a kinetic reaction included in the code but we assume that the reaction is occurring at fast rate in the 120-140$^o C$ window. This is slightly smaller than the 100-150$^o C$ reported in the literature and it is aimed at roughly accounting for a slow kinetic at lower temperatures and the lack of reactant (smectite) left at higher temperatures. During this phase of fast reaction, fluids





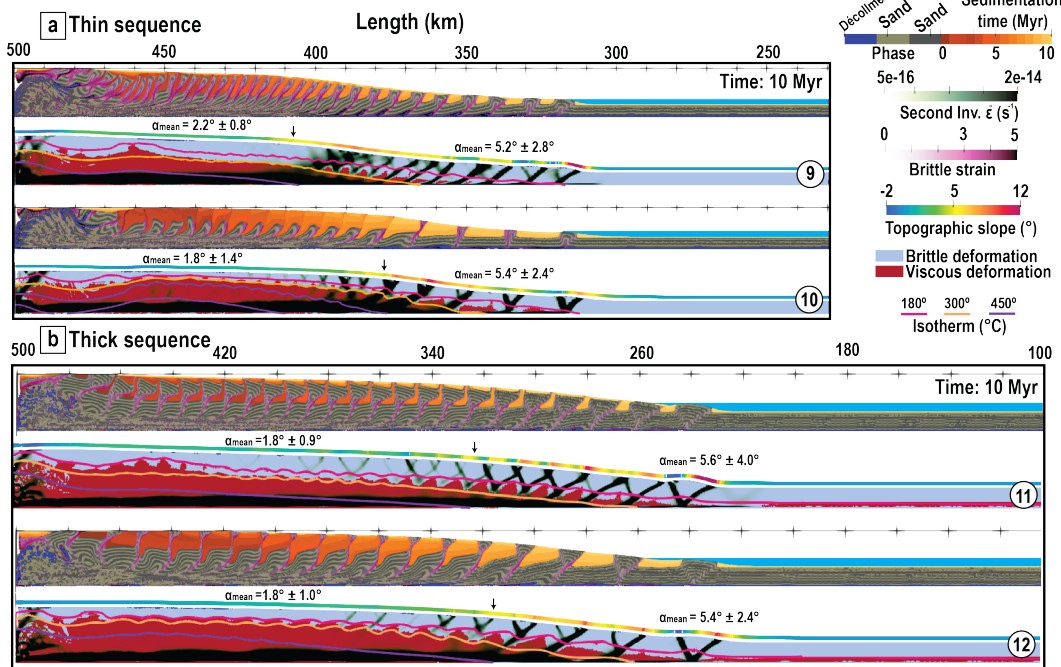

**Figure 6.** High temperature gradient (25 $^oC/km$) after 10 Myr for **a.** thin sequence and **b.** thick sequence. **M9** and **M11** are with normal erosion and sedimentation rate whereas **M10** and **M12** indicate high sedimentation rate. Color codes and experiment parameters are given in Figure 2 and Tables 1 and 2.

are released in the clay rich décollement of low permeability permitting to build strong local fluid over pressure (Bekins et al., 1994; Lanson et al., 2009). The code does not explicitly include fluids, but these overpressures are reflected by an effective friction angle of 0.1$^o$ within the reaction temperature window. The evolution of friction with temperature in the décollement layer is reported in Figure 7.

Figure 8 shows the structural evolution of simulation **M14**, identical to the reference model **M1** but accounting for the smectite-illite transition.

After 1 Ma, by the formation of conjugate shear bands, the wedge tries to reach its critical state in accordance with basal friction of $\phi_b = 5^o$. Since the base quickly reaches 120 $^oC$, the frontal wedge is extremely narrow, and followed by a flat area induced by the drop of friction ($\phi_b = 0.1^o$), as expected from CTT.

At 3.8 Ma, the flat area is highly extended ($x = 380 - 470 km$). The onset of this transition serves as the root to a shallow splay fault accompanied by a back-thrust.

At 6.3 Ma, as the shortening goes on, the temperature rises due to burial. Once the décollement reaches 140 $^oC$, the basal friction increases to 10$^o$, leading to the onset of a larger topographic slope. Once the décollement reaches 180 $^oC$, some patches of viscous deformation appear in between active thrusts (at $x = 440 km$). As observed in model **M1**, a segment of the





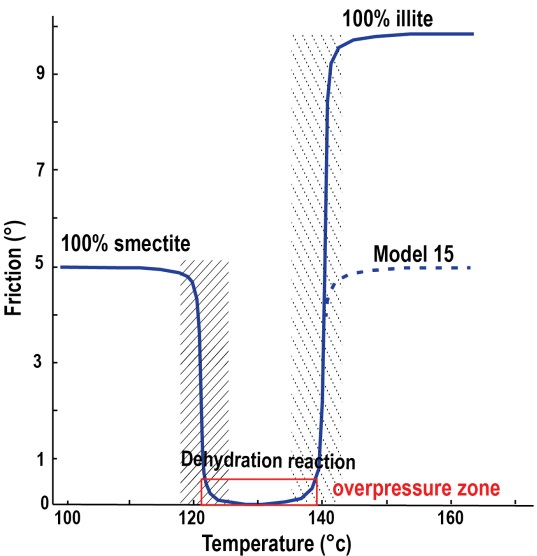

**Figure 7.** Evolution of friction with temperature with our parametrisation of smectite-illite transition. The dashed line is for the evolution of friction in simulation M15 which serves as comparison with M1 at the brittle-ductile transition zone.

high topographic slope, formed by a low angle thrust and its backthrusts, separates the flat segment from the warmer distributed part of the wedge.

At 10 Ma, the wedge is characterized by four distinct segments: a narrow frontal critical taper, a weakly deformed zone with a flat topographic slope associated with the smectite-illite transition (120 to 140 $^oC$), a high topographic slope corresponding to the brittle-ductile transition formed by a low angle thrust and its backthrusts rooting on the ductile décollement, and, finally, a second flat segment corresponding to the ductile wedge once the décollement reaches a temperature of 450 $^oC$. The brittle-ductile transition segment shows a larger topographic slope than the reference model **M1** (Figure 3), because of the higher basal

friction reached once the dehydration reaction finishes ($\phi_b = 10^o$). From 10 to 15 Ma, the warm and flat part of the ductile wedge grows in length like in model **M1**.

   Figure 8b illustrates that shear heating tends to increase the extent of the flat dehydration segment by reducing both the size of the critical taper located at the toe of the wedge and the lateral extent of the steep segment located at the brittle-ductile transition. Figure 8c demonstrates that the increase of slope of the brittle-ductile segment in model **M14** as compared to model

**M1** is indeed related to the increase in basal friction up to $10^o$ after the dehydration reaction.

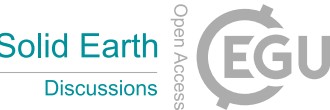

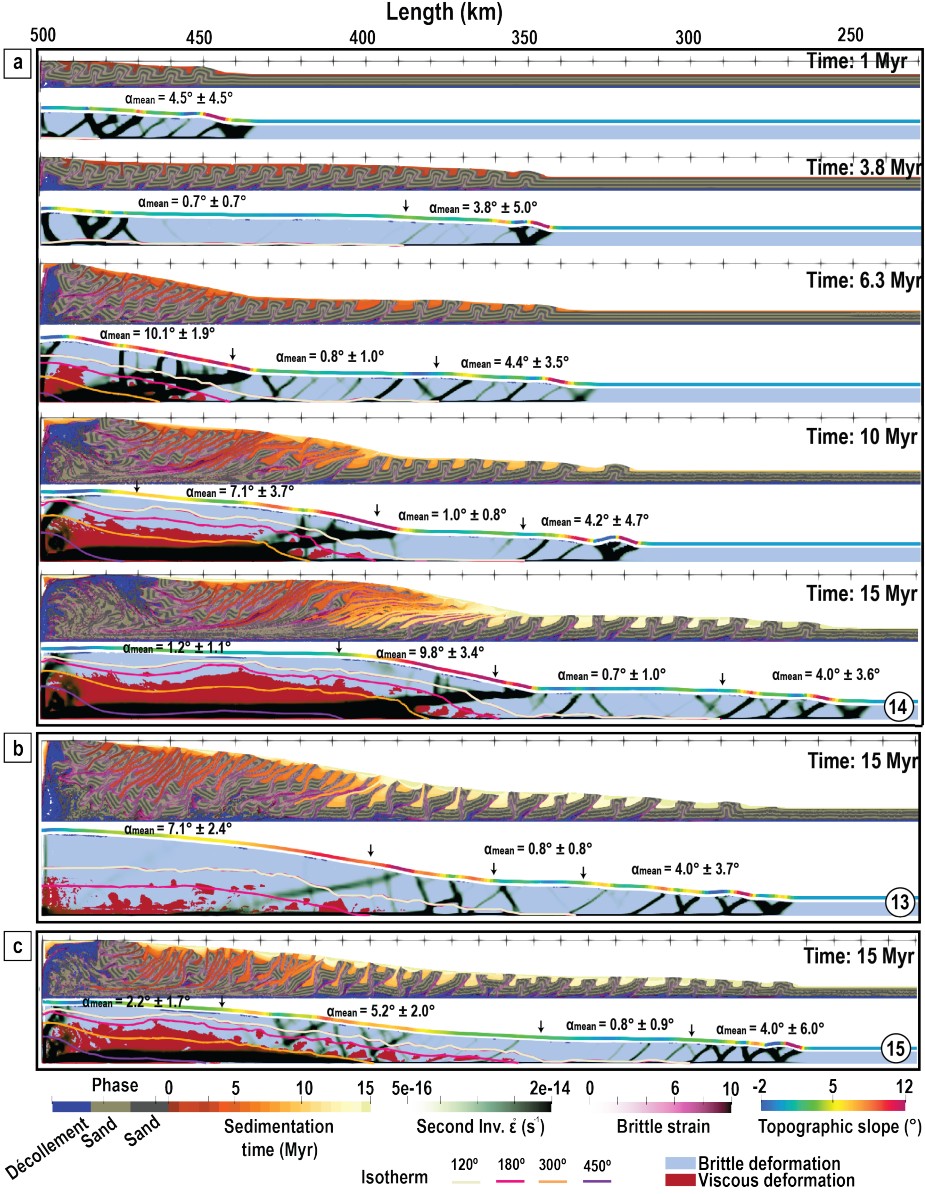

**Figure 8. a.** 15 Myr of time evolution of simulation $M14$ with dehydration reaction and shear heating . **b.** cross-sections at 15 Myr for similar simulation without shear heating $M13$. **c.** cross-sections at 15 Myr for similar simulation $M15$ with $5^o$ friction instead of $10^o$ friction after the end of dehydration reaction.





## 4 Discussion

### 4.1 Slopes and modes of deformation

The thermomechanical model provides an opportunity to investigate the specific variations of topographic slope of an accretionary prism, supported by the CTT analysis (Davis et al., 1983). Our results show that very simple models of accretionary prism lead to the formation of four different structural zones which corresponds to three different type phases of deformation related to transitions in the rheology.

The frontal brittle part of the wedge is characterized by an imbricated zone and active in-sequence thrusts faults ahead. The décollement and the above sequence behave plastically (Figure 9a). The topographic slope created in this section is controlled by the basal and average internal frictions and is consistent with the CTT predictions (Figure 9b and c, blue star).

The presence of a smectite-illite transition (dehydration reaction) leads to a segment characterized by a flat topographic slope and little internal deformation, in between the frontal brittle wedge and the brittle-ductile transition (Figure 9a). This flat segment appears during the early stage of the accretionary prism formation. As the wedge shortens and grows, the temperature increases at the back due to burial and the wedge becomes thicker and warmer, reducing this frontal flat segment.

The brittle-viscous transition zone is characterized by out-of-sequence thrusts and backthrusts with high internal deformation (Figure 9a). This part forms a steeper topography slope than the brittle part. A careful examination of the behavior of this segment reveals that the décollement remains brittle, but the above sequence has entered the viscous phase (Figure 9a). By plotting the surface slope on the critical taper diagram, we notice that this part is consistent with a critical taper of a lower internal friction angle (Figure 9b, red star).

The viscous part presents an approximate flat zone without effective internal deformation. Décollement and above sequence deform viscously. The topographic slope is again consistent with the critical taper theory, considering that a viscous décollement is equivalent to a brittle décollement of extremely low friction. Therefore, the increase of the topographic slope between the brittle and viscous segments results from an equivalent decrease of internal friction rather than an increase of basal friction.

### 4.2 Comparision with exhumed accretionary prism

Based on the models, we can interpret that the rocks exhumed at the back of the models have been through 3 main phases of deformation through time. The first phase D1 is purely brittle and corresponds to the onset of accretion at the toe of the accretionary prism. The passive transport through the dehydration flat segment does not cause important deformation. D1 is therefore overprinted by a phase D2, which corresponds to the start development of a low grade metamorphic foliation as a result of penetrative horizontal shortening. This change in material behavior at depth, from brittle to viscous, due to an increase of temperature at depth, favors the activation of brittle backthrusts and causes the steepening of the slope. In an accretionary prism with large basal heat flow, or thick incoming sedimentary, the sequence D2 appears closer to the toe of the prism and might replace D1. Warmer models then develop a phase in which deformation is partitioned between ductile simple shear at depth (D3) and distributed shortening at the surface (more penetrative version of D2). Once the horizontal ductile shear zone corresponding to D3 has formed, it branches on a shallow dipping splay fault accompanied by a very vertical back-thrust. The





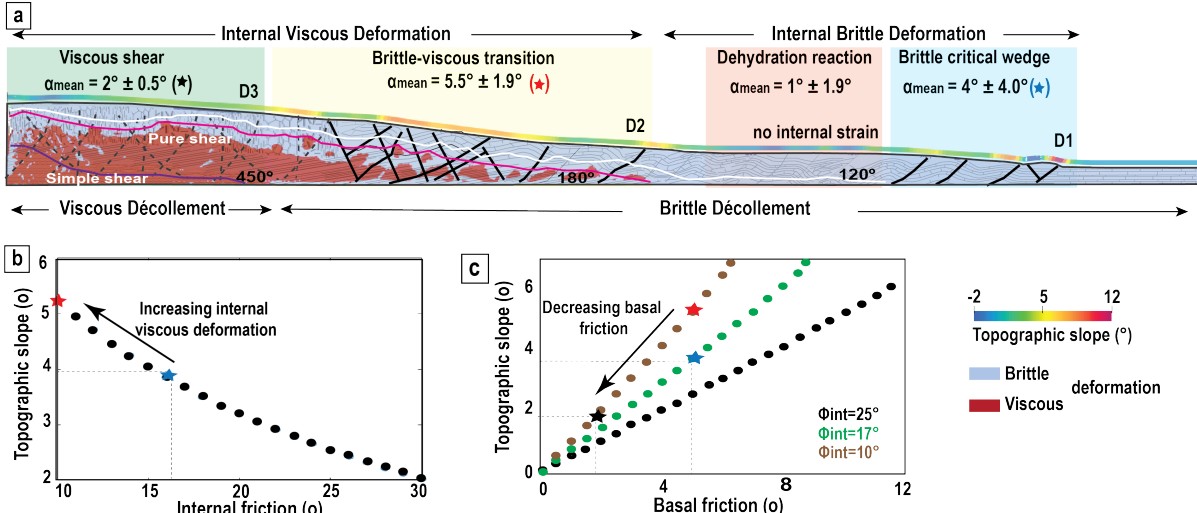

**Figure 9. a.** Proposed model for the mature brittle ductile wedge which forms three distinct segments; pure brittle wedge with a rather constant slope predicted by the CTT at the front (blue star), low but non zero topographic slope close to the backstop corresponds to viscous deformation (black star) and, larger topographic slope in between these segments as a result of the brittle ductile transition (red star and yellow rectangle). **b.** Topographic slope versus internal friction for $\phi_{basal}$=5°. **c.** Basal friction versus topographic slope for $\phi_{int}$ 10, 17 and 25°.

three phases of deformation recorded by the rocks exhumed at the back of the models along a normal fault correspond to the

340 different phases recorded in exhumed accretionary prism like the Shimanto Belt (Raimbourg et al., 2014), although this paper would classify our D1 and D2 as more or less localised deformation related to frontal accretion and in our case the high grade ductile foliation D3 would show a more marked asymmetry.

### 4.3 Forearcs basins

Every phase of deformation is accompanied by different types of sedimentary basins. Indeed, while phase D1 is accompanied

345 by piggy-back basins which length depends on the sedimentation rate (e.g., Figure 4), D2 is accompanied with the formation of trench slope basins discordant on the early piggy-back basins, and the onset of D3 pinpoints the start of activity of the splay fault and its back-thrust. These structures isolate the ductile part of the prism, where forearc sediments can accumulate within small basins in discordance on the sediments accreted during D1 and D2, from the brittle part of the prism. The splay fault and its conjugate high angle back-thrust serve as a current backstop for the brittle part before being incorporated into the ductile

350 forearc part of the prism as accretion continues.

Backthrusting between the imbricated segment and forearc basins have been described along various accretionary margins of high sedimentation rate (Silver and Reed, 1988). Along the Sumatra subduction zone, well-known for its high sedimentation rate and high thermal gradient (Chlieh et al., 2008), the slope break predicted by our model is visible on seismic images of





northern (Chauhan et al., 2009) and southwestern (Singh et al., 2010) Sumatra, and a clear backthrust has been imaged down to
7 s (15 km) (Chauhan et al., 2009). Backthrusts have also been imaged along the Antilles subduction zone, in particular along
the Barbados region, again known for its high sedimentation rate psilver1988backthrusting, laigle2013along. A backthrust
is also well documented in New Zealand (Barnes and Nicol, 2004) but has been interpreted as resulting from a change in
basal friction. Noda (2016) reviews a number of other compressive accretional margins which all seem to develop an active
backthrust at the edge of the forearc basin like in our models.

However, most of these seismic studies place the splay fault and its back-thrust at the limit of the continental crust which
posits, in a way, a stable position through time, at least relative to the upper plate. In our simulations, the location of the splay
fault is given a more dynamic nature as it corresponds more or less to the $450^oC$ isotherm (Figure 10a). This isotherm at 15 km
depth corresponds to greenschist facies metamorphic conditions. Using typical sediments composition, perplex software yields
typical continental crust seismic attributes with 5.5 > Vp > 5 km/s and Vs  2.5 km/s. With volcanoclastic sediments, larger
velocities are expected for similar metamorphic conditions. As a result, seismic refraction investigation of active margin would
definitely identify this part of the models as a continental crust or former arc crust. Using the location of the splay fault in warm,
compressional accretionary contexts like Southern Sumatra (Figure 10b) and Lesser Antilles (Figure 10c), we propose that what
is typically interpreted as attenuated continental or arc crust could also well mark the location of the brittle-ductile transition.
According to our models, along with accretionary prisms of little seismic activity, the forearc basin should correlate with the
fully viscous domain at least along high sedimentation rate and high thermal gradient compressive accretionary margins.

## 4.4   Up and down-dip end of the seismogenic zone

We here confirm that the smectite-illite transition produces a flat segment that can explain, for young or cold accretionary
complexes, the observed correlation between deep-sea terraces or fore-arc basins with large subduction earthquakes (Song
and Simons, 2003; Wells et al., 2003). In such specific contexts, the flat segment would thus underline the up-dip limit of the
seismogenic zone. Along seismically active accretionary prisms, an increase of topographic slope and decrease in apparent
geodetic coupling are interpreted as the down-dip limit of the seismogenic zone (Cubas et al., 2013). In the first case, the rise in
topographic slope indicates an increase of effective basal friction which corresponds to the down-dip end of dominant seismic
slip (Cubas et al., 2013; Pajang et al., 2021). In the second case, geodetic deformation at velocities that differs from subducting
plate velocities is generally interpreted as a lack of coupling on the plate interface based on elastic models (Perfettini et al.,
2010; Chlieh et al., 2008). Here we show that the brittle-ductile transition corresponds to the onset of internal distributed
viscous deformation in between brittle structures in the accretionary prism above a brittle décollement. This causes a decreased
geodetic coupling and an increased apparent basal friction but does not imply any modification in the interface frictional
properties or deformation mode. This questions both the search for field analogs of the down-dip end of the seismogenic zone
and the interpretation of seismic coupling in terms of locked/creeping décollement.

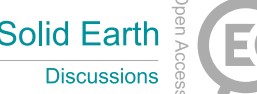

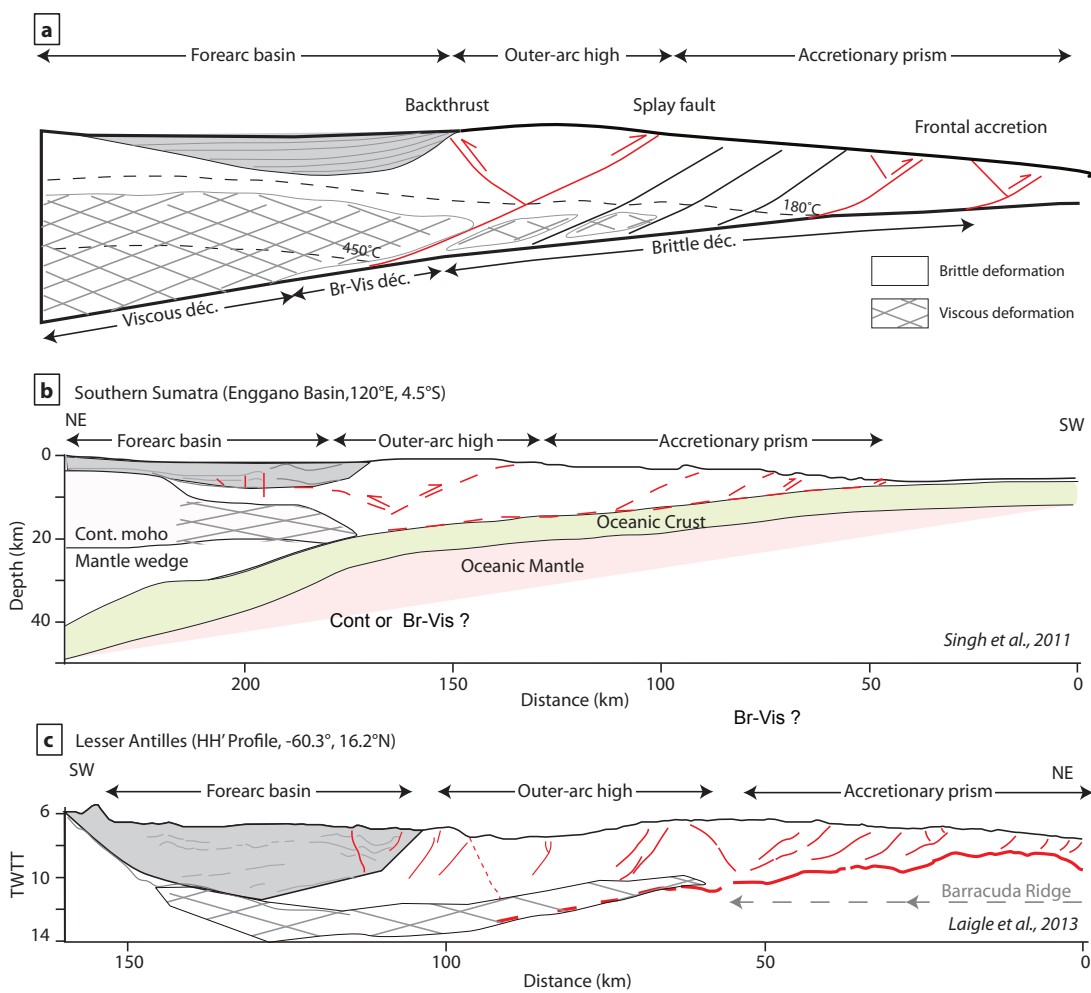

**Figure 10. a.** Structure of accretionary wedge and forearc basin modified from Noda (2016) based on our modelling results for the geometry of isotherms and the distribution of brittle and viscous strain. Schematic cross-sections of **b.** Southern Sumatra based on Singh et al. (2011) **c.** Lesser antilles based on Laigle et al. (2013) with indication of the possible location of the brittle ductile transition based on our model and the observed geometry of faults.





### 4.5 Limitations and perspectives

Our simple models permit us to understand first order mechanisms in forearc formation, but improvements are still needed to better understand how the brittle-ductile transition affects the formation of splay faults providing an alternative model to the formation of forearc basins as well as an alternative origin for the forearc crust. We find that, the drop in friction between the ductile and brittle part of the accretionary prism and smectite to illite dehydration reaction is not sufficient to explain the

normal faults observed along some accretionary margins like Chile or Makran Cubas et al. (2013); Pajang et al. (2021). This result leads us to conclude that the normal faults arise from phenomena we neglected in our simplified approach. This includes:

- the effect of heterogeneities in the subducting plate,

- the effect of elastic deformation during the seismic cycle and/or

- the over simplification of the bottom boundary conditions which does not allow an increase in taper angle with burial
like in Beaumont et al. (1992); Ruh (2020).

We plan on testing these other hypotheses in the future and posit that improving the bottom mechanical boundary conditions will also permit performing more accurate simulations of the sedimentation in the forearc basins by keeping them below sea level.

## 5 Conclusions

Despite the simplicity and limitations of our simulations, their results are sufficient to propose a new model for the interpretation of changes in topographic slope at compressive accretionary margins, at least, warm margins with large sedimentary supply. Using only the topographic gradient, our model distinguished four segments, which corresponds to 3 modes of deformations observed in exhumed accretionary complexes:

- brittle décollement and internal deformation at the toe where the topographic slope respects the CTT;

- a flat area with little internal deformation in young accretionary prisms, if a dehydration reaction is considered;

- brittle décollement and viscous internal deformation of fault-bounded blocks, where the topographic slope is in excess compared to the CTT and these large slopes should not be interpreted as the downdip limit of the seismogenic zone;

- viscous décollement and backthrust bounded blocks in the most internal part where the topographic slope is close to zero.

The most important finding is that the onset of internal viscous deformation in fault-bounded blocks increases the topographic slope of the accretionary complex independently of the basal friction.

Comparing the simulations results with natural cases, we show that this anomalous topography related to the brittle-ductile transition is analogue to the forearc high in a compressional accretionary prism. It is indeed the location of an active back-thrust and splay fault system rooting on the viscous channel that forms in the internal part where the basal temperature reaches



$450^{o}C$. The location of this viscous channel, which is active in greenschist facies conditions, corresponds to portions of seismic sections that are often interpreted as an attenuated continental crust or former arc crust. Our model provides, we believe, a valid alternative interpretation which has the advantage to explain why the forearc crust is always thinner than the continental crust on seismic sections but also why the warm subduction segments are representative of our mature stage simulation, such as South Sumatra or the Lesser Antilles are considered aseismic.

*Code availability.* The version of ptatin2d and the input files used in this contribution are archived following FAIR principle at https://doi.org/10.5281/zenodo.4911354. Input files are located in published_inputs/Pajang2021_SE

*Video supplement.* all simulations are available as movies archived at https://doi.org/10.5281/zenodo.5599365

*Author contributions.* SP Designed the simulation, run the simulation prepared the figure and wrote the paper, LLP implemented the heat flow bc, LLP and NC, participated to the interpretation of the simulation, application to natural cases and writing the paper

425   .

*Competing interests.* Authors declare no competing interest

*Acknowledgements.* Authors thanks Tiphaine Larvet for computing seismic velocity with perplex.





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
