# Peer review of "The topographic signature of temperature controlled rheological transitions in an accretionary prism"

_Solid Earth, 2021_

## Author Comment (AC2)

**Revisions:**

**Dear Professor Patrice Rey:**

At first, we would like to thank editorial handling by Professor Patrice Rey and the reviewers for their valuable comments that have helped us to improve the content of the manuscript.

The comments of each reviewer are now addressed in more detail below.

The modifications text in blue color are based on first reviewer's points and in green color based on second reviewer's points.

The number of lines has changed due to the addition or subtraction of some content and paragraphs compared to the original manuscript file.

We hope that these revisions and improvements of the manuscript make it suitable for a wide audience and ready for publication in the international journal of Solid Earth.

Best wishes,

Sepideh Pajang on behalf of the co-authors.

**Dear first Reviewer (se-2021-135)**

Dear first reviewer thanks for your accuracy on our manuscript. Your comments were really valuable and we have corrected the manuscript in regard of your remarks. The manuscript improved by implementing your comments. Again, thanks for your guidance, and taking the time to read our manuscript.

**General comments:**

**1.1 Reviewer's comment:** I found the spatial distribution of viscous and brittle deformation in the brittle-viscous transition zone very interesting. I think that the Authors could provide some more clarity for their interpretations of the observed relationships. Specifically, most models show low strain islands of sand to deform viscously at temperatures between 300°C and 180°C. These islands of viscous deformation are bounded by faults characterised by brittle deformation. Moreover, deformation in the décollement remains brittle while part of the overlying sequence deforms viscously at lower temperatures. I am wondering how the Authors interpret these rheological relationships.

**Authors' reply:** This is due to large variation in strain rate, as well as brittle softening, low strain island deforms but at rates that are much smaller than brittle faults which are weaker. None the less as brittle soft faults rotate and become less well oriented and as the temperature rises making the low strain island weaker, viscous deformation occurs in between brittle faults.

**1.2 Reviewer's comment:** Is viscous deformation of quartz-dominated lithologies expected and what could cause the onset of viscous deformation at temperatures between 300°C and 180°C? If all sandstone in the sandstone sequences has the same mechanical properties, what causes the concurrent viscous and brittle deformation at a specific depth and temperature?

**Authors' reply:** Brittle softening in eq. 10.

After equation 10 which we added :

Friction and cohesion drop "permits to former faults to remain brittle where undeformed rocks creep viscously. The decollement is exempt of softening both to facilitate the comparison with CTT and because it is considered originally frictionally weak."

**2. Reviewer's comment:** In D2 deformation phase occurring in the brittle-viscous transition zone, back-thrusts are attributed to the change in the material behaviour from brittle to viscous with depth (e.g., lines 333-334). In Figure 9a, however, the backthrusts are shown to be rooted within or at the top of the brittle décollement. The Authors could consider providing some clarity and additional explanation regarding the processes or conditions that favour the formation of back-thrusts at this part of the wedge.

**Authors' reply:** Conjugate thrusts (fore- and back-thusts) initially form at the transition of the brittle and viscous behavior to compensate different slip rates.

The backthrust is favored due to the large topographic slope behind: it is easier to scrap off the material at the front rather than to uplift a large wedge with a high topographic slope.

This is added to the text: " As temperature increases with burial, material behaviour in between faults which deforms with lower strain rates changes from brittle to ductile decreasing the effective friction coefficient of the bulk. This in turns causes the steepening of the slope which favors the activation of brittle backthrusts, that scrap off the frontal wedge rather than forethrusts that would have to uplift a lot of material."

**3. Reviewer's comment:** I found the titles of subsections 3.2 and 3.3 not very informative. I do not have any good suggestions of how the titles could be improved, but the Authors could rethink these titles.

**Authors' reply:** They are changed to: "3.2 Time evolution of reference model

3.3 sensitivity to shear heating, erosion and thickness of incoming sediments"

**4. Reviewer's comment:** I found the colour schemes used in the models hard to follow and interpret. Less so in the "current state figure" of each model, where the type of deformation (brittle versus viscous), strain rate, and topographic slope are presented. In the "finite strain figure", I found it hard to discriminate between the colouring describing the amount of brittle strain and the colouring for lithology / sedimentation time. For example, it hard to determine whether there is any brittle strain accumulated in the deposited sediments (i.e. deposited after 0 Ma). I am not sure how easy would be to fix this issue. I leave it to the discretion of the Authors to decide whether they wish to address this issue.

**Authors' reply:** The reviewer is right that the color palette for brittle strain would not show well in the sediments, but we actually did not plot the brittle strain in the sediments for this reason. One can see, from the strain rate that sediments do deform, and the brittle/localized deformation in the sediments is outlined by the deformation of their bedding.

As we already tested different color code, the one used in the manuscript was the best, we did not change the figure but we added this precision in the post processing part.

**5. Reviewer's comment:** The use of the term "strain" is not clear. The text refers to finite strain (e.g., lines 145, 147), the figure legends suggest that it is the brittle strain mapped on the models. Observing the models, however, brittle strain appears in domains where the wedge deforms purely viscously. Also, from a rheological perspective, it might be interesting to show the spatial distribution of stress magnitude in the evolving wedge.

**Authors' reply:** we indeed map the brittle strain only on the figure, total strain can be deduced from the geometry of the incoming sediments. It is normal to have brittle strain in the viscously deforming area, because brittle strain is acquired during D1, but the brittle shear zone can be further deformed viscously during D3 if viscous creep is weaker mechanism than brittle yielding.

It is added to the text " For each simulation, we show the finite brittle strain in the rocks and the strain rate of the current state. The total strain (brittle and viscous) can be deduced from the geometry of the incoming sediments (grey and black originally horizontal)."

**6. Reviewer's comment:** I am wondering if the Authors have explored the relationship between the slope of the isotherms and the topographic slope in their models.

**Authors' reply:** We did not, but they are in general almost parallel to the topography due to diffusion (poisson problem with dirchlet bc at top and Neumann at bottom) with some wiggles which corresponds to thrust activity (advection flux) and variations of diffusivity in the sediments. Dirichlet at top force the flux to be normal to the surface.

**Specific comments:**

**1.Reviewer's comment:** Line 11, Please explain what aspect of the brittle-ductile transition results in increase of the topographic slope (e.g., depth?).

**Authors' reply:** "by decreasing internal friction" is added to the text.

**2. Reviewer's comment:** Line 13, Please change to: "Our models, therefore, imply ….".

**Authors' reply:** Thanks for your comment, it is done.

**3. Reviewer's comment:** Lines 37-38, "…, which stability field is controlled mainly by temperature" – Please rephrase this part of the sentence. Something seems to be missing.

**Authors' reply:** True, modified as "Clay minerals are phyllosilicate-hydrated and their stability field is mainly controlled by temperature"

**4. Reviewer's comment:** Line 49, Change "complex" to "complexes".

**Authors' reply:** Thanks for your comment, done.

**5. Reviewer's comment:** Line 50, Please change "rich" to "reach".

**Authors' reply:** Thanks for your comment, done.

**6. Reviewer's comment:** Lines 58-66, It is hard to follow the content of this sentence, primarily because of the large number of citations. Also, the "While" in the beginning of the sentence does not fit to how the sentence evolves. It seems that something is missing. I suggest rewriting the sentence.

**Authors' reply:** True, 'While' is omitted and number of citations reduced by using "e.g.,"

**7. Reviewer's comment:** Line 68, "…how the introduction temperature evolution…" – Please check the sentence. Something seems to be missing.

**Authors' reply:** "of" was missing, now added in the text.

**8. Reviewer's comment:** Line 73-74, "We briefly discuss internal deformation the morphology of the wedge and its potential seismic behavior." – Please check the sentence. It may need some rewording.

**Authors' reply:** We have improved the sentence as " We briefly discuss the internal deformation and the morphology of the wedge and also its potential seismic behavior"

**9. Reviewer's comment:** Line 127-128, It is not clear to me to what "respectively" refers in this sentence. Does it refer to the two different initial thicknesses of the model? If so, please make it clearer.

**Authors' reply:** Thanks for this point, It is clearer now: " for 4 and 7.5 km thickness"

**10. Reviewer's comment:** Line 144, Do you mean on the right of the panels?

**Authors' reply:** Thanks for your comment, done.

**11. Reviewer's comment:** Line 145, Please explain what you mean by "current state". Also, in the text you mention "finite strain" while in the figures you report "brittle strain". Are these considered the same, in your descriptions? Please explain.

**Authors' reply:** For each simulation, we show the finite brittle strain and the strain rate of the current state. The total strain (brittle and viscous) can be deduced from the geometry of the incoming sediments. More explanation is available in comment 5 (General comments).

**12. Reviewer's comment:** Line 183, Change "trust" to "thrust".

**Authors' reply:** Thanks for your comment, it is done.

**13. Reviewer's comment:** Line 193, "…which corresponds to the brittle-ductile transition." – The brittle-ductile transition is present in almost the half length of the model. Do the Authors mean that the topographic slope corresponds to the slope of the 300°C isotherm, taken to correspond to the brittle-ductile transition? If not, please explain as this outcome is important but not clearly presented.

**Authors' reply:** No, higher topographic slope corresponds to the brittle-ductile transition which is a range between 180 to 450 °C isotherms.

We expended the sentence because it is very important point of the paper and it was also unclear to reviewer 2. "A mature brittle-ductile wedge forms three distinct segments that can be distinguished based on topographic slope. The back segment,

close to the backstop where the decollement is viscous display a rather low but non zero topographic slope. The third segment, at the toe where the wedge is purely brittle, displays a CTT predicted slope. In between, where both brittle and ductile deformation co-exist within the wedge while the decollement is still brittle, a central segment displays a distinctively larger topographic slope than predicted by CTT. We refer to that segment as the brittle-ductile transition segment of the wedge."

**14. Reviewer's comment:** Line 200, Do the Authors mean faults, instead of shear bands and shear zones? Deformation seems to be entirely brittle after 1 Myr.

**Authors' reply:** In numerical models, faults are brittle shear bands because they are not discrete. We added brittle to make it clear.

**15. Reviewer's comment:** Line 213: Please elaborate on how strain rate shows information about the thickness of the wedge.

**Authors' reply:** Strain rate indeed does not inform on the thickness of the wedge but the text mentions the thickness of the shear zone not the thickness of the wedge. So, we did not change anything.

**16. Reviewer's comment:** Lines 223-224, I would agree that the brittle-ductile transition in the wedge seems to reach some sort of steady state configuration, but in my view, this takes place between 15 and 20 Myr.

**Authors' reply:** Thanks for your comment, done.

**17. Reviewer's comment:** Lines 226-229, The text does not flow very well in these lines. Please consider rewriting. For example:

"This phase corresponds to crossing the zone…." - It is not clear to which zone refers, and what crosses the zone.

"…whether they were incorporated in the ramp or not…" - It is not clear what is meant by "they".

"…before being exhumed for large temperature" – again, it is not clear to me what the Authors try to say here.

**Authors' reply:** we changed to:

"This phase of deformation corresponds to the moment at which the incoming sediments are incorporated to the second segment of the wedge where the topographic slope is larger than CTT predictions."

**18. Reviewer's comment:** Line 241, Maybe "run" instead of "ran"?

**Authors' reply:** Thanks for your comment, done.

**19. Reviewer's comment:** Line 248-252**,** This sentence is long and quite complicate. Also, the last part of the sentence does not flow well. The Authors could consider simplifying the sentence.

**Authors' reply:** The section is modified with your point.

"Actually, exhumation is reached under two conditions in our models, with large erosion coefficients, i.e. M4 and M6, and in presence of shear heating. The peak metamorphic temperature of rocks exhumed at the back-stop is compatible with thermochronometry studies in stationary accretionary prism like Taiwan (Suppe et al., 1981; Willett and Brandon, 2002). Its samples are exhumed to the surface by rock uplift to compensate for the mass lost via erosion (Fuller et al., 2006) and they have experienced temperatures in excess of 300–365oC but below 440oC e.g., (Lo and Onstott, 1995; Fuller et al., 2006)."

**20. Reviewer's comment:** Line 255, Please provide the number of kilometres along the models in Figure 5, where the out-of-sequence thrusts appear.

**Authors' reply:** The distance in km added to the text.

**21. Reviewer's comment:** Lines 263-265, This sentence needs to be simplified or broken in two, in my view. It is a long, dense sentence, and it does not read smoothly.

**Authors' reply:** They are broken and smoothed as "In absence of heat production and large vertical advective terms, the temperature is more or less proportional to the depth and thermal gradient in the models. Therefore, in experiments with thick sequences (M5 and M6 in Figure 5 and M7 and M8 in Figure 4) or in models with larger imposed basal gradient (M9, 10, 11, 12 in Figure 6), the brittle-ductile transition is reached earlier**.**"

**22. Reviewer's comment:** Line 302**,** "…by reducing both the size of the critical taper…" – Do the Authors mean the size of the critical taper angle? If so, it would be useful to provide the values for models M14 and M13.

Moreover, there are significant differences between models M14 and M13, potentially even more striking from the two mentioned in the text. The Authors could expand on this aspect.

**Authors' reply:** No, we meant the size of the brittle wedge with a basal friction of 5 degrees.

A sentence added at the end of the paragraph "As described above, without shear heating, the flat plateau above the viscous decollement hardly develops."

**23. Reviewer's comment:** Line 317-318**,** in Figure 8a, the length of the frontal flat segment decreases between 3.8 and 6.3 Myr. After 6.3 Myr, I do not see any significant change in its length. Especially at 15 Myr, the length of the Frontal flat segment seems to me larger compared to 10 Myr. In case I am wrong, it might be preferable if the Authors describe quantitatively the change of length over time.

**Authors' reply:** The reviewer is correct, the evolution of the size to the flat segment with time is a bit more complex than the elusive sentence we put in the text. In a first draft, we had a long paragraph about the evolution of the flat segment with time and this is remnant of it left in the discussions. To say the truth, the time evolution is quite complex and if we are correct and this segment is the seismogenic zone, this would deserve a paper on its own because it depends on the thickness of the incoming sediments, the exact temperature of dehydration etc. So, we just, removed that sentence from the text.

**24. Reviewer's comment:** Line 328, Change "Comparision" to "Comparison".

**Authors' reply:** Thanks for your comment, it is done.

**25. Reviewer's comment:** Line 332, "…which corresponds to the start development of…" – Please consider rephrasing.

**Authors' reply:** The section is modified as "The D1 is therefore overprinted by the phase D2, which is the start of a low-grade metamorphic foliation as a result of penetrative horizontal shortening."

**26. Reviewer's comment:** Line 335, "…or thick incoming sedimentary,…" – Something seems to be missing here.

**Authors' reply:** Changed to "thick incoming sediments".

**27. Reviewer's comment:** Line 338, "…by a very vertical back-thrust." – Change to "a vertical back-thrust" or "a steeply-dipping back-thrust".

**Authors' reply:** Thanks for your comment, done.

**28. Reviewer's comment:** Line 343, Change to "Forearc basins".

**Authors' reply:** Thanks for your comment, done.

**29. Reviewer's comment:** Line 356, Something has gone wrong in this line. Does not make sense.

**Authors' reply:** Thanks for your comment, done.

**30. Reviewer's comment:** Line 362, "…it corresponds more or less to the $450^{\circ}$C isotherm" – I think a more accurate description would be that the splay fault roots at the location of the $450^{\circ}$C isotherm along the décollement. Or something similar.

**Authors' reply:** Thanks for your comment, done.

**31. Reviewer's comment:** Lines 365-366, Is it possible to provide examples of the velocities recorded at the base of the spay fault, by seismic studies of active margins? It would be useful for comparison with the velocities you report here.

**Authors' reply**: For Sumatra, Chauhan et al, 2010 (Sumatra) >=6; Kopp et al., 2013 (Chile) >5; Kopp et al., 2002 (Java) > 5.5.

"with Vp >= 5" added to the text.

**32. Reviewer's comment:** Lines 369-370, For comparison reasons, the Authors may wish to add a figure callout to one of their models where the forearc basin forms on top of the viscous shear domain.

**Authors' reply:** *"(Fig. 9a)"* added.

**33. Reviewer's comment:** Line 381-383, Please be more specific what is meant by "This" in the beginning of each sentence.

**Authors' reply:** This refers to " internal deformation", is added in the text.

**34. Reviewer's comment:** Fig. 9a, An explanation for the isotherm lines is missing from the legend.

**Authors' reply:** True, because we mentioned the temperature beside each isotherm.

**Dear Dr. Guillaume Duclaux**

Dear Dr. Duclaux thanks for your accuracy on our manuscript. Your comments were really valuable and we have corrected the manuscript in regard of your remarks. The manuscript improved by implementing your comments. Again, thanks for your guidance, and taking the time to read our manuscript.

**1. Reviewer's comment:** My main concern is related to the model flat base and its impact on the dynamics of the prism itself. In the classical CTT approach the prism internally evolves to maintain a balance between a plunging basal décollement (plunging landward in natural cases) and the surface topography. Because here the basal decollement is flat (parallel to the base of the model box), the simulated surface topography should be overestimated, and not in direct agreement with observations. I understand this is an actual limitation of the model, but it should be further discussed as it might impact the whole topographic slope analysis presented by the authors. Still, I remain fairly convinced with the authors study, as although the absolute topographic slopes predicted in their models might be wrong, the relative change in the slopes linked to brittle-ductile transition or metamorphic reactions should still exist.

**Authors' reply:** Our main concern in the discussion is focused on varying beta with the weight of the column using flexural bc as presented in Ruh et al (2020). The code can actually easily handle constant landward slope by including the slope in the gravity vector, we just decided not to vary too many parameters in this study which focuses on self-consistent brittle ductile transition and which main novelty resides in including heat flow BC's at the base of the model.

The reviewer is correct, the topography is not correct in the model, but the relative change of slope is not affected by a constant landward slope, and actually the slope them-self are affected according to our modified CTT which account for a drop of effective internal friction at the BDT.

None the less, we realize now that including the reference model with a constant landward basal slope in the discussion will strengthen our argument so we did add it together with a paragraph of discussion.
"Since we run simulations with basal slope b= 0, according to CTT the simulated topographic slope might be overestimated compared with natural examples. We thus run the reference simulation with a basal slope of 2 degrees. We found the same segmentation but with slightly lower topographic slopes as expected from the CTT (supplementary movie M16)."

**2. Reviewer's comment:** Shear heating is a very important factor in this study as it is the sole heat production term accounted for in the energy equation. Now, the unit for the heat production by shear heating doesn't make sense to me... according to Eq. 4 it is in Pa.s?? How? Some additional explanations are necessary. Could the authors please clarify this in the methodology section?

**Authors' reply:** Oops sorry for the missing time derivative on the strain!! And thank you for picking up that one.

Using Einstein notation, the corrected equation 4 in 2D plane strain is:

$$H = \tau_{ij}\dot{\varepsilon}_{ij} = 2\eta\left(\dot{\varepsilon}_{xx}^2 + 2\dot{\varepsilon}_{xy}^2 + \dot{\varepsilon}_{yy}^2\right)$$

We chose not to include the dimensional analysis in the paper because it is standard, once the dot is added on the epsilon but we add it here if this was really the point of your remark:

$$Pa => N \cdot m^{-2}$$
$$J => N \cdot m => Pa \cdot m^3$$
$$W => J \cdot s^{-1}$$

So

$$Pa.s^{-1} \; is \; W \cdot m^3$$

if you divide it by $\rho(\mathrm{kg} \cdot m^{-3})Cp(K/kg/J)$ you obtain$(\mathrm{k} \cdot s^{-1})$ which is the dimension of equation 3.

**3. Reviewer's comment:** A vertical scale must be added to each figure presenting the model results (Fig. 2 to 7, and 9). There is a very important vertical exaggeration in these figures. Understanding this is critical to compare the models with natural cases and make this work directly usable to others.

**Authors' reply:** There is no vertical exaggeration in the figures, Now, "1:1 scale" is added in the captions.

**4.1 Reviewer's comment:** Boundary conditions (BCs) are of prime importance in numerical models. The fixed left wall certainly has a strong influence on the development of the normal fault described by the authors, and the exhumation pattern of the dome visible in various models. Have you considered, or tested, alternative BCs?

**Authors' reply:** Yes, we did. A free slip, causes a big overturned fold, but it is not consistent with CTT nor sandbox experiments, nor nature unless the backstop is covered with salt.

The goal in this paper is to stay as close as possible to sandbox experiments in order to really analyse the role of BDT which cannot be capture in these experiments.

**4.2 Reviewer's comment:** To be clear I'm not asking the authors to run additional models here, but the importance of the fixed left wall on the dynamics of the ductile region of the prism should be pointed out in the discussion, if not conceptually explored.

**Authors' reply:** We added a sentence about the fact that the normal fault at the back is most probably an inevitable boundary effect.

A normal fault forms near the backstop "(which could be probably an inevitable boundary effect)" and ….

**Minor comments**:

**1.Reviewer's comment:** l. 31: replace "along" with "within"

**Authors' reply:** Thanks for your comment, it is done.

**2. Reviewer's comment:** l. 50: reach instead of rich

**Authors' reply:** Thanks for your comment, done.

**3. Reviewer's comment:** l. 68: missing "of" --> the introduction of temperature

**Authors' reply:** Thanks for your comment, done.

**4. Reviewer's comment:** l. 91 and 95: to be consistent with the rest of the manuscript please use an upper case "E" for Eq. and Eqs

**Authors' reply:** Thanks, it is done.

**5. Reviewer's comment:** l. 91: shear heating, see comment above.

**Authors' reply:** It is done.

**6. Reviewer's comment:** l. 127-128: please rephrase the sentence about mesh elements as it isn't very clear. It reads as if all models had 2 independent meshes, one for the shale unit, and one for the sediments above... But, as I understand it some experiments have 16 elements vertically, others have 24 (ny in Table 2).

**Authors' reply:** True, corrected.

**7. Reviewer's comment:** l. 139: remove the "/" before (

**Authors' reply:** Thanks for your comment, done.

**8. Reviewer's comment:** l. 165: "k" font should be in italic

**Authors' reply:** Thanks for your comment, done.

**9. Reviewer's comment:** l. 191: I would start a new sentence at the start of this line with "The mature [...]"

**Authors' reply:** It is done.

**10. Reviewer's comment:** l. 193: "topographic slope which corresponds to the brittle-ductile transition" --> ok, but could you please be more specific? Indeed, the brittle-ductile transition is present in M1 everywhere there is viscous deformation, on the left side of the model, at median depth between the surface and the bottom of the box.

**Authors' reply**: We already changed and clarified that part according to the reviewer 1 and we are sorry, we did not realize that what we call the brittle-ductile transition in

our everyday Jargon corresponds to the anomalous topographic slope segment and not the isotherm 300°C or whatever it is as a function of strain rate. Thank you both.

Now: "A mature brittle-ductile wedge forms three distinct segments that can be distinguished based on topographic slope. The back segment, close to the backstop where the decollement is viscous displays a rather low but non zero topographic slope. The third segment, at the toe where the wedge is purely brittle, displays a CTT predicted slope. In between, where both brittle and ductile deformation co-exist within the wedge while the decollement is still brittle, a central segment displays a distinctively larger topographic slope than predicted by CTT. We refer to that segment as the brittle-ductile transition segment of the wedge."

**11. Reviewer's comment:** l. 219: The vertical partitioning between simple and pure shear is not so clear to me. The base is indeed dominated by simple shear, but as far as I can see the top is barely deforming at that stage, except near the dome that forms next to the backstop. Could you provide some additional arguments/evidences for the pure shear deformation please across the model? Or is it limited to the region near the backstop?

**Authors' reply:** Deformation is indeed limited but not inexistant outside of the backstop. You can inferred from the discordant mini-basins at the top of the sedimentary sequence. We refer to it as pure shear because it seems to be accommodated by conjugate shear bands.

**12. Reviewer's comment:** l. 320: I would suggest adding 'only' in "slope than the brittle-only part".

**Authors' reply:** Thanks for your comment, done.

**13. Reviewer's comment:** l. 323-333: I understand the D2 metamorphic foliation should be subvertical, is that correct? Please describe the superimposed fabric orientation in the context of the model. Is this in agreement with observations in forearc wedges?

**Authors' reply:** D2 is the start of a low-grade metamorphic foliation as a result of penetrative horizontal shortening. This part has been corrected.

Yes, it is in agreement with the Shimanto belt as discussed at the end of the paragraph.

**14. Reviewer's comment:** l. 355: The 7s are supposedly TWT (two-way time)? Please precise this here.

**Authors' reply:** TWT is added

**15. Reviewer's comment:** + l. 356: references formatting is incorrect. Please fix that.

**Authors' reply:** Thanks for your comment, done.

**16. Reviewer's comment:** l. 367: I would recommend the authors look into Vérati et al (2018 --> https://doi.org/10.1016/j.lithos.2018.08.005) paper about the development of low grade metamorphic foliation in the volcanic pile of the Lesser Antilles arc. Although it is not directly the sedimentary prism, this work seems very relevant to this study.

**Authors' reply:** Thank you for pointing out this paper to us.

We added " In the lesser Antilles arc, Vérati et al. 2018 have documented distributed deformation by pressure solution occurring at 300°C/ 4-5 km depth condition within the brittle accretionary prism which could corresponds to exhumed remnants of D2."

**17. Reviewer's comment:** l 376, 378 and 382: It's probably me, but I'm not familiar with the term "geodetic coupling" or "geodetic deformation". Could you please explain those terms?

**Authors' reply:** Geodetic coupling: subduction megathrust locked during the interseismic period as revealed by GSNN.

Based on your comment, now is added to the text "the seismogenic zone (part of the megathrust geodetically coupled i.e., locked during the interseismic period)".

**18. Reviewer's comment:** Fig 4: for consistency with the main text please replace high sedimentation with surface diffusion.

**Authors' reply:** Thanks for your comment, done**.**

**19. Reviewer's comment:** Fig 8 b) and c): Do not use "cross-sections". A cross-section from a 2d model would be a 1D line... These are the "models".

**Authors' reply:** Thanks for your comment, done**.**

Thanks again for your precision and follow up on our manuscript.